# Polyarteritis Nodosa: Old Disease, New Etiologies

**DOI:** 10.3390/ijms242316668

**Published:** 2023-11-23

**Authors:** Louis Wolff, Alice Horisberger, Laura Moi, Maria P. Karampetsou, Denis Comte

**Affiliations:** 1Department of Internal Medicine, Hôpital Universitaire de Bruxelles (H.U.B.), Université Libre de Bruxelles (ULB), 1050 Brussels, Belgium; louis.wolff@ulb.be; 2Department of Medicine, Division of Rheumatology, Inflammation, and Immunity, Brigham and Women’s Hospital, Harvard Medical School, Boston, MA 02115, USA; ahorisberger@bwh.harvard.edu; 3Department of Medicine, Division of Immunology and Allergy, Lausanne University Hospital, University of Lausanne, 1011 Lausanne, Switzerland; 4Immunology and Allergology, Institut Central des Hôpitaux, Valais Hospital, 1951 Sion, Switzerland; laura.moi@hopitalvs.ch; 5Rheumatology Private Practice, 11635 Athens, Greece; mkarampet@gmail.com; 6Department of Medicine, Division of Internal Medicine, Lausanne University Hospital, University of Lausanne, 1005 Lausanne, Switzerland

**Keywords:** PAN, polyarteritis nodosa, panarteritis nodosa, monogenic, VEXAS, DADA2

## Abstract

Polyarteritis nodosa (PAN), also known as panarteritis nodosa, represents a form of necrotizing vasculitis that predominantly affects medium-sized vessels, although it is not restricted to them and can also involve smaller vessels. The clinical presentation is heterogeneous and characterized by a significant number of patients exhibiting general symptoms, including asthenia, fever, and unintended weight loss. Although PAN can involve virtually any organ, it preferentially affects the skin, nervous system, and the gastrointestinal tract. Orchitis is a rare but specific manifestation of PAN. The absence of granulomas, glomerulonephritis, and anti-neutrophil cytoplasmic antibodies serves to distinguish PAN from other types of vasculitis. Major complications consist of hemorrhagic and thrombotic events occurring in mesenteric, cardiac, cerebral, and renal systems. Historically, PAN was frequently linked to hepatitis B virus (HBV) infection, but this association has dramatically changed in recent years due to declining HBV prevalence. Current epidemiological research often identifies a connection between PAN and genetic syndromes as well as neoplasia. This article provides a comprehensive review of PAN, specifically focusing on the progression of its clinical manifestations over time.

## 1. Introduction

PAN was first described in 1866 by Kussmaul and Maier. They reported an “intermittent nodular appearance affecting arteries throughout the body, sparing large vessels (the aorta and its branches), small vessels (arterioles, capillaries, and venules) and pulmonary vessels” [1]. The distinct “pearl necklace” pattern led to the naming of this condition as polyarteritis nodosa. In 1952, pathologist Pearl Zeek laid out the first classification of PAN, distinguishing between hypersensitivity vasculitis, allergic vasculitis, PAN, and temporal arteritis, now, respectively, known as urticarial vasculitis, eosinophilic granulomatosis with polyangiitis (Churg–Strauss syndrome), PAN, and giant cell arteritis (Horton’s disease) [2]. A pivotal development came in 1982 when autoantibodies directed against neutrophil cytoplasm antigens (ANCA) were identified in eight patients with clinical characteristics of vasculitis, introducing ANCA-associated vasculitis (AAV) as a distinct category of vasculitis separate from PAN [3]. The most recent definition of PAN came from the 2012 Chapel Hill Consensus Conference (CHCC) where the disease was described as “necrotizing arteritis of the medium or small arteries without glomerulonephritis or vasculitis of arterioles, capillaries or venules and without ANCA” [4,5,6]. Over the past two decades, the medical community’s understanding of PAN has significantly evolved. While initially described as either primary vasculitis or Hepatitis B virus (HBV)-related, the current understanding recognizes PAN as a disease often secondary to genetic syndromes and malignant hematologic disorders [7]. In this review, we examine the evolution of our understanding of PAN over the years by providing key aspects of epidemiology, pathophysiology, and clinical presentations and discussing how these translate into the current therapeutic approach.

## 2. Epidemiology

The prevalence of PAN varies greatly across different countries, ranging from 2 to 31 per million inhabitants in Europe [8,9]. There are notable north–south and seasonal gradients in the occurrence of the disease. Historically, PAN was frequently linked to HBV infection. With the advent of its vaccine and the enhancement of public health measures in response to the AIDS crisis, the incidence of PAN has seen a significant decline, turning it from one of the most common vasculitis in the 1990s to one of the least common today [8,10]. In addition, the improvement in laboratory techniques for the detection of ANCA in the 1980s led to the reclassification of certain vasculitis that were initially diagnosed as PAN [11]. The challenge in estimating the overall prevalence of PAN arises from a combination of factors, including the absence of serum markers, heterogeneous classification criteria, and a variety of predisposing genetic and environmental factors.

In the cohort from Pagnoux et al., which included 348 patients enrolled between 1963 and 2005, the mean age was 51 ± 17 years old with a male to female ratio of 1.7, and no ethnic predominance was observed [12]. A more recent cohort from Rohmer et al., comprising 197 patients identified with PAN from 2005 to 2019, had a mean age of 53.6 ± 18 years and a male predominance of 1.5 [13]. In addition, over 300 cases of a PAN-like disease in children and young adults mainly presenting with fever, skin ulcers, early-onset stroke, peripheral neuropathy, hypogammaglobulinemia, cytopenias, and elevated acute phase reactants who were found positive for adenosine deaminase 2 mutations (DADA2 disease) have been described in the literature since 2014. In a study of 118 patients with idiopathic adult PAN, 4.3% were found to have biallelic pathogenic variants of ADA2, and several others had less clearly significant ADA2 variants [14]. Finally, PAN may be one of the manifestations, or even the initial manifestation, of a rare autoinflammatory disease known as VEXAS syndrome. VEXAS syndrome was initially described in 2020 and is caused by somatic mutations in methionine-41 of *UBA1*, the major E1 enzyme that initiates ubiquitylation. Large exome sequencing efforts have identified the prevalence of VEXAS syndrome-associated pathogenic variants at approximately 1:4000 in men and 1:26,000 in women over the age of 50 [15]. In the initial report describing VEXAS, 3 out of 12 patients met the criteria for PAN. Since then, another nine cases of medium-vessel vasculitis have been attributed to VEXAS syndrome [16,17].

## 3. Physiopathology

PAN is typically characterized by a segmental, necrotizing, and transmural inflammation, predominantly involving small- to medium-sized arteries, although any arterial size could theoretically be susceptible. The disease most commonly impacts the visceral and muscular arteries, including their branches. In patient biopsies, it is common to observe co-existing lesions of diverse stages of inflammation and scarring within a single sample [18]. Because of the arterial inflammatory process, fibrinoid necrosis may develop, leading to the formation of microaneurysms. Over time, these complications can progress to chronic stages, characterized by fibrous scarring and vascular aneurysms, which can rupture and lead to severe bleeding [19,20]. During the acute phase, the cellular infiltrate, composed of macrophages, T lymphocytes, neutrophils, and eosinophils, is generally observed in the tunica media but can also invade the tunica interna and tunica externa [21]. One distinctive feature of PAN, compared to other vasculitis, is the absence of granulomas. This disease is also characterized by the coexistence of different stages of vascular inflammations at the same time.

The pathophysiology of PAN, not yet fully understood, may vary depending on the disease’s specific etiology. Serum cytokine profile analysis in PAN patients has revealed an elevation in interferon-alpha (IFN-⍺), interleukine-2 (IL-2), tumor necrosis factor-α (TNF⍺), and IL-1-ß compared to healthy individuals and those with granulomatosis with polyangiitis (GPA) [22]. Immunohistochemical studies of muscle and nerve biopsies from patients showed the presence of macrophages (41%) and mostly CD4+ T lymphocytes (41%) [19,23]. Most of these studies, however, primarily focus on PAN associated with HBV.

Viral infections remain a common trigger of PAN and should be excluded in all cases. In PAN associated with HBV, the HBs antigen is responsible for the formation of immune complexes [24,25], as suggested by animal models of hepatitis B antigen-associated PAN, which show an accumulation of immune complexes in blood vessels [26,27]. Hepatitis C virus (HCV) has also been linked to PAN, with HCV-associated PAN tending to present more severe and acute symptoms [28,29]. However, this only concerns 5% of patients with PAN, and the distinction with cryoglobulinemic vasculitis can sometimes be challenging [30]. HIV infection has been associated with PAN, though HIV-associated PAN is generally less aggressive than HBV-associated PAN. The classical manifestation is mononeuritis multiplex and can occur at any stage of HIV infection [31]. Although parvovirus B19 has been associated with PAN, a study using PCR tests found no higher prevalence of this infection in people with PAN compared to those without [32,33,34,35]. 

More recently, vasculitis has been associated with COVID-19 infection, but to date no cases of PAN have been reported [36,37]. COVID-19 vaccines have been associated with PAN manifestations [38,39,40]. Like other vasculitides, PAN can be induced by the use of certain drugs, such as minocycline [41]. The association between PAN and neoplasia is well established, especially for hematological malignancies, such as hairy T cell leukemia, or, more recently, myelodysplastic syndrome (MDS) [42,43,44,45,46]. In a study by Roupie et al., out of 70 patients with MDS and vasculitis, 9% presented with PAN. MDS is associated with a pro-inflammatory state in 10–30%, which also tends to present with autoimmune and inflammatory disorders. MDS and certain chronic inflammatory diseases share common genetic markers (such as HLA-B27) and polymorphisms (such as IL-1) [47,48]. 

More recently, genetic forms of PAN have been described. In the early 2000s, cases of PAN-like vasculitis were described in patients with Familial Mediterranean Fever (FMF) [49,50]. In a nationwide study in Turkey, PAN prevalence in patients with FMF was 0.9% [51]. FMF is caused by mutations in the MEFV gene that encodes for pyrin/marenostrin, which result in unregulated production of IL-1, leading to recurring inflammation, fever, and, sometimes, autoimmune manifestations [52]. Patients with PAN associated with FMF present a higher incidence of perirenal hemorrhages and elevated levels of inflammation [49,50,51,53]. Another condition related to genetic forms of PAN is STING-associated vasculopathy, with onset in infancy (SAVI), which is a type I interferonopathy. This condition is caused by mutations in the TMEM173 gene that induce the inflammation of endothelial cells in children. It often presents PAN-like symptoms in affected children [54]. A monogenic syndrome resulting from a deficiency in Adenosine Deaminase 2 (DAD2) has been described in familial cases of necrotizing vasculitis that resemble PAN [55]. Since 2014, over 60 bi-allelic loss-of-function mutations in the ADA2 gene have been documented [56,57]. Vascular inflammation in DAD2 patients is believed to be caused by an imbalance in macrophages, favoring the M1 type over the M2 type. To date, more than 200 cases of this condition have been recorded [58,59]. In 2020, Beck et al. published a cohort study of 25 men exhibiting a somatic mutation affecting methionine-41 (p.Met41) in the *UBA1* gene. Located on the X chromosome, this gene encodes for a critical enzyme involved in the initiation of ubiquitination. The syndrome associated with this mutation is known as VEXAS, an acronym for Vacuoles, E1 enzyme, X-linked, Autoinflammatory, and Somatic. Although PAN-like features were initially reported in 12% of these patients, more recent studies suggest a lower incidence [16,60] (Figure 1).

## 4. Clinical Manifestations

Signs and symptoms of PAN result from damage to the vascular walls, potentially affecting all organs. This section provides an overview of organ systems that can be impacted in PAN patients. Unless otherwise stated, the percentages and specifics of the manifestations come from the cohorts shown in Table 1.

### 4.1. General Symptoms (85–93%)

General, non-specific symptoms such as asthenia, fever, weight loss, myalgia, and arthralgia are frequently the initial symptoms of PAN. They are present in over 9 out of 10 patients.

### 4.2. Neurologic (59–79%)

Neurologic manifestations occur in more than two-thirds of patients, most commonly as motor and sensory mononeuritis multiplex of the peripheral nerves [12,64]. Peripheral neuropathy is typically distal, asymmetric, and can be rapid-onset, often associated with localized skin edema. Of note, deep sensation is rarely affected. Foot drop, an important and disabling complication, may be the initial presentation [65]. Cranial nerves are affected in less than 1% of patients. Central manifestations such as strokes occur in 2 to 10% of patients, typically in the later stages of the disease [64,66,67]. Compared with classic PAN, DADA2 patients more frequently present with central neurologic manifestations, particularly stroke, usually at a young age.

### 4.3. Cutaneous (50–59%)

Skin lesions, including nodules, purpura, necrotic ulcers, and livedo reticularis, are present in half of the patients [68]. In cases of cutaneous manifestations suggestive of vasculitis, a skin biopsy is recommended. The biopsy should be deep enough to include the dermal layer where medium-sized arteries are located. A subset of PAN, known as cutaneous PAN (CPAN), is confined to the skin and requires different management [69,70].

### 4.4. Renal (15–75%)

Kidney involvement is typically characterized by stenosis and aneurysm, primarily affecting renal and interlobar arteries and less frequently affecting the smaller arcuate and interlobular arteries [71,72]. This can manifest as hypertension (up to 35% of patients), micro or macro hematuria, mild proteinuria, and renal infarct [12,72,73]. Despite up to 75% of patients experiencing renal involvement, renal insufficiency occurs in only 15% of cases [12,74]. Glomerular involvement is infrequent [75]. Severe complications such as ruptured aneurysm and spontaneous perirenal hemorrhage, which require embolization or nephrectomy, are infrequent but can be life-threatening [76,77,78,79].

### 4.5. Gastrointestinal (22 to 38%)

Gastrointestinal manifestations of PAN are common, affecting up to 50% of patients. Abdominal pain is reported by one out of three patients [80,81]. Vascular inflammation in mesenteric arteries can be severe, leading to intestinal ischemia, perforation, and hemorrhage [82,83]. Reports have also shown gallbladder involvement, malabsorption with loss of weight, and pancreatitis in patients with PAN [84,85]. In rare but severe instances, hepatic aneurysm can occur, potentially triggering acute liver failure and resulting in high mortality [86,87,88,89,90]. Gastrointestinal manifestations are associated with a poor prognosis; the mortality rate is around 25% for patients with such involvement [12,82]. This association with high mortality is supported by a retrospective study from 1988, which found that digestive complications contributed to the death of 16% of PAN patients [91,92]. Diagnosing mesenteric arterial involvement in PAN can be challenging, and conventional angiography may be valuable, especially in patients with minimal clinical evidence of extraintestinal manifestations.

### 4.6. Genital (15 to 17%)

Manifestations, such as testicular pain, with or without orchitis, have been described as potential symptoms specific to PAN, although similar symptoms have been noted in Behcet’s patients. Testicular biopsy can be useful for diagnosis [93]. Interestingly, there are reports of ovarian artery dissection associated with PAN in women [94]. 

### 4.7. Cardiovascular (7 to 78%)

Cardiac involvement predominantly affects the myocardium due to coronary artery vasculitis. The left anterior descending, circumflex branches, and right coronary arteries are most affected. Pericarditis is relatively rare, often resulting from pre-existing myocardial involvement [95,96,97]. Myocardial infarction due to coronary infarction is also unusual [98]. Celiac artery involvement and new-onset hypertension are potential risk factors for coronary involvement [99]. Heart failure often presents during the initial stages of the disease. Hypertrophic cardiomyopathy, which may be a result of uncontrolled hypertension, could trigger serious conditions like ventricular tachycardia and syncope. Mild diffuse interstitial myocarditis can be caused by focal necrosis [100]. Among pediatric patients with PAN, cases of hemopericardium have been described [101,102,103]. In terms of vascular manifestations, large vessels can be affected due to necrosis of the vasa vasorum [104]. Symptoms of arterial claudication can be indicative of stenosis or ischemia in the lower extremities [105,106,107].

### 4.8. Other Manifestations

Ophthalmic complications, including retinal vasculitis, are observed in PAN [108,109,110]. Unlike other forms of vasculitis, such as granulomatosis with polyangiitis (GPA), eosinophilic granulomatosis with polyangiitis (EGPA), or microscopic polyangiitis (MPA), pulmonary lesions are notably absent in PAN. However, an autopsy study revealed bronchial artery damage in 7 out of 10 patients, despite the absence of symptoms [111]. Muscular manifestations in PAN can vary from nonspecific myalgia to paresis, and muscle biopsy shows inflammation related to PAN in up to 50% of cases [112]. 

VEXAS syndrome presents general symptoms like fever or weight loss in 96% of patients, skin manifestations such as neutrophilic dermatosis or tender plaques (84%), pulmonary infiltrates (49%), chondritis (36%), and deep vein thrombosis (35%) [62]. Hematologic manifestations include macrocytic anemia (96%) and vacuoles in bone marrow myeloid and erythroid cells [16]. The vascular manifestations of VEXAS mimic small to large vessel vasculitis [16,63,113]. In the inaugural study of 25 VEXAS patients, 12% were diagnosed with PAN [57]. A recent literature review showed that among nine cases of medium vessel vasculitis found, all were men with macrocytic anemia and skin lesions, six of whom had passed away prior to the article’s publication [112]. Patients with DADA2 exhibit vasculitis, immunodeficiency, and hematological manifestations. Vasculitis appears as mucocutaneous manifestations in 75% of cases, including livedo reticularis (50%), PAN-like skin lesions with non-granulomatous necrotizing inflammation of medium-sized arteries (34%), digital necrosis (22%), nodules (14%), Raynaud’s phenomenon (8%), and aphthous ulcers (7%). Neurological manifestations occur in 51% of cases and may include ischemic strokes (27%), cranial nerve palsy (27%), hemorrhagic strokes (12%), and polyneuropathy (9%). General symptoms such as fever-elevated erythrocyte sedimentation rate or CRP are present in half of the patients. Immunodeficiency manifests as hypogammaglobulinemia (22%), low IgM (18%), low IgA (12%), and infections (20%). Viral infections (11%) are more frequent than bacterial ones (7%). Hematological diseases manifest as anemia (13%), neutropenia (7%), and thrombocytopenia (6%) due to bone marrow failure or autoimmune cytopenia. Lymphoproliferative symptoms (32%), including splenomegaly and lymphadenopathy, are common in patients with DADA2. Most symptoms and signs (85%) occur before the age of 12 [63,114] (see Table 1).

## 5. Treatments

The treatment recommendations for PAN are primarily based on weak empirical evidence and are often drawn from recommendations for other forms of vasculitis, with modifications according to the disease severity. Mild PAN, characterized by non-life- or organ-threatening manifestations like constitutional symptoms, arthritis, or skin lesions, is differentiated from moderate to severe PAN, which involves more severe complications, such as arterial stenosis—particularly those involving the renal arteries and aorta—and ischemic complications that affect the heart, peripheral nervous system, and gastrointestinal system. To aid in risk stratification, the 1996 version of the Five Factor Score (FFS) can be used. This score assigns +1 point for each of the following: proteinuria greater > 1 g/day, serum creatinine > 140 µmol/L, cardiomyopathy, severe gastrointestinal involvement, and CNS involvement [115,116]. 

Treatment for mild PAN (FFS of 0) may include glucocorticoids (GC) only. The clinical benefit of supplementing glucocorticoids with an immunosuppressive agent is not definitively established, but it could potentially offset the high 40% relapse rate and function as a steroid-sparing strategy. Guidelines, however, show divergence in recommendations. The French protocol typically prescribes glucocorticoids as a standalone treatment, introducing immunosuppressants such as methotrexate or azathioprine only in instances of resistance or intolerance. In contrast, the ACR’s 2021 guidelines advocate for a combined approach from the beginning, recommending the incorporation of azathioprine (administered orally at 2–3 mg/kg/day) or methotrexate (preferably given subcutaneously at 0.3 mg/kg/week) with glucocorticoids. Moderate to severe PAN (FFS > 0) is treated with intravenous (IV) GC in conjunction with an immunosuppressive agent, preferably cyclophosphamide. The start of treatment marks the induction phase, lasting 3 to 6 months, aimed at achieving disease remission, defined by the American College of Rheumatology (ACR) as a complete absence of clinical manifestations, with or without immunosuppressive treatment. Initial treatment strategies recommend starting with at least 1 mg/kg/day of prednisone equivalent, capped at 60 mg/day. In patients with severe manifestations requiring rapid intervention, IV boluses of methylprednisolone are recommended. If remission is incomplete, the duration of cyclophosphamide therapy may be extended, although it is recommended not to exceed a period of 6 months given its potential toxicity [115,116]. 

Alternative therapies, including rituximab, mycophenolate mofetil, tocilizumab, anti-TNF alpha, JAK inhibitors, IV immunoglobulins, or plasma exchange, have not been well studied and their application is only reserved for certain refractory or relapsed patients [34,117,118,119,120,121,122,123,124,125]. A recent European retrospective study analyzed 42 patients treated for relapsed and/or refractory PAN. Tocilizumab, anti-TNF alpha, and rituximab achieved complete remission in 50%, 40%, and 33% of cases, respectively, with comparable safety profiles. These biotherapies may become first-line treatments in the future, but more data are needed [119]. The induction phase is followed by the maintenance phase, with the objective of preventing relapse. Patients treated with cyclophosphamide with complete remission may be switched to azathioprine or methotrexate for 12 to 18 months [115,116]. In secondary forms of PAN, the therapeutic approach focuses on the underlying etiology. In the context of HBV-associated PAN, antiviral therapy is used as the primary intervention. In severe cases, management with GC and plasma exchange may be considered [126]. When PAN is concomitant with MDS, interventions targeting the MDS are often effective in attenuating the vasculitic manifestations [43]. From this perspective, Mekinian et al. showed that azacytidine successfully treated autoimmune manifestations in 9 out of 11 patients with MDS [127]. In the case of DADA2-associated PAN, numerous treatments have been explored (azathioprine, cyclosporine, tacrolimus, cyclophosphamide, and methotrexate) with mitigated results. The ACR 2021 guidelines have now approved the use of steroids and anti-TNF alpha-agents (etanercept, infliximab, or adalimumab) following demonstration of their efficacy in PAN associated with DADA2 [115]. Hematopoietic stem cell transplantation (HSCT) has been reported to treat cytopenia [63]. Regarding VEXAS syndrome, several drugs have been tested with mixed results. GC in combination with azacytidine (possibly supplemented by HSCT) seems most effective for patients with MDS features. For those without myelodysplasia, JAK inhibitors or tocilizumab may be suitable [128,129,130]. Finally, vasculitis associated with primary immunodeficiency can be managed with biotherapies, HSCT, or IV immunoglobulin therapy [55,131].

## 6. Conclusions

Over the past two decades, the understanding of PAN has significantly evolved. Though PAN was previously considered primarily as either idiopathic or HBV-related, the landscape now encompasses a broader spectrum that includes PAN associated with infections, paraneoplastic syndromes, and newly recognized classifications such as DADA2, interferonopathies, and VEXAS syndrome. Identification of the underlying causes of PAN is critical for directing treatment strategies. While conventional immunosuppressants, such as cyclophosphamide, are often the standard of care for primary PAN, secondary forms may respond better to more specific agents: anti-TNF alpha for DADA2, JAK inhibitors for VEXAS, and targeted MDS treatment for PAN associated with MDS or allogeneic stem cell transplantation.

## Figures and Tables

**Figure 1 ijms-24-16668-f001:**
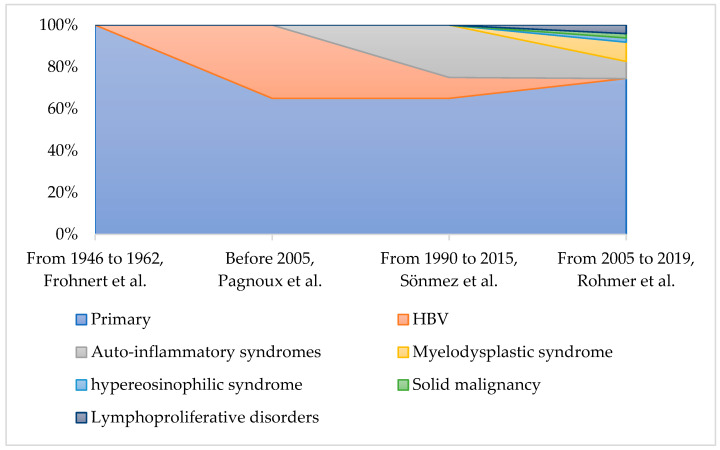
Comparison of various etiologies of PAN across different cohorts over distinct time periods. Adapted from Refs. [7,10,12,61].

**Table 1 ijms-24-16668-t001:** Characteristics of PAN patients reported in different cohorts (PNP: peripheral neuropathy, CNS: central nervous system, FFS: Five Factor Score, 1996). Results are expressed as percentages.

Characteristics	Pagnoux et al. (1963 to 2005) [12]	Sönmez et al. (1990 to 2015) [10]	Rohmer et al. (2005 to 2019) [12]	Georgin-Lavialle et al. (VEXAS) [62]	Meyts et al. (ADA2) [63]
General symptoms	93.1		85	95.7	50
Fever	63.8	53.7	54	64.6	
Loss of weight	69.5	53.7	50	54.5	
Myalgia	58.6	46.2	50		
All cutaneous	49.7	67.2	59	83.6	75
Nodules	17.2				14
Purpura	22.1				
Livedo	16.7	17.9			50
Panniculitis			7.5	12.9	
Renal	50.6	47.7	20	9.5	
Hematuria	15.2				
Proteinuria	21.6				
Hypertension	34.8	41.7			21
Orchitis	17	14.9	16		4
Neurologic	79.0	43.2	59		
PNP	74.1			5.2	9
Mononeuritis	70.7			2.6	
CNS	4.6				53
Digestive	37.9	22.3	28	13.8	33
Abdominal pain	35.6	37.3		8.6	12
Bleeding	3.4			0.9	
Perforation	4.3			0.9	2
Cardiovascular	22.4		39		
Pericarditis	5.5			4.3	
Distal necrosis	6.3	13.5			22
Thrombo-embolism				35.3	
Ophthalmic	8.6			40.5	
Retinal vasculitis	4.3				
Pulmonary		2.9	8	49.1	
Cough	5.7				
Lung infiltrate	3.4			40.5	
Pleural effusion	3.4			9.5	
Chondritis				36.2	
Arthralgia	48.9	58.2		28.4	
Arthritis		17.9			

## Data Availability

No new data were created.

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
