# Peer review of "Polyarteritis Nodosa: Old Disease, New Etiologies"

_ijms, 2023, doi:10.3390/ijms242316668_

Round 1
Reviewer 1 Report
Comments and Suggestions for Authors
Dear Authors, I have read your manuscript with interest.
The current manuscript titled: "Polyarteritis nodosa reviewed in 2023: old disease, new etiologies" represents an important analysis of evolving field of Rheumatology and Immunology.
In my opinion, these are the adjustments which should be made to increase the value of your manuscript:
1. It is recommended to exclude the 2023 year from the manuscript title so that the article does not lose relevance after a certain period of time.
2. In Introduction chapter, please, add the review aim.
3. In chapter 2, please, add more detailed information about PAN epidemiology.
4. Please, change “Covid-19” to “COVID-19”.
5. Please, add abbreviation for “COVID-19” and “SARS-CoV-2”.
6. It is recommended to improve the Figure 1 quality and change the font to the same with the general manuscript text.
7. Please, add detailed information about PAN differential diagnosis and future perspective PAN treatment.
8. It is recommended to study and consider this recent holistic article about PAN https://doi.org/10.3390/medicina59061162.
9. The Conclusions section is very extensive. Please, highlight the practical implications of this review and its relevance to real clinical practice.
10. The manuscript contains some punctuation errors, please revise the text.
Comments on the Quality of English LanguageMinor editing of English language required
Reviewer 2 Report
Comments and Suggestions for Authors
The authors are to be congratulated for presenting a well-referenced and up to date review of this rare vasculitis condition. This is very timely, given the recent publication of the clinical update from the French Vasculitis Study Group (Rohmer 2023). When compared to some other 'recent' reviews of PAN this review is significantly more comprehensive and informative. The authors have also summarised case reports where biologic therapies have been used.
The review is well organised and referenced. I have one minor issue: Table 1 lists the manifestations of PAN v VEXAS v DADA2: since only 12% of VEXAS patients have PAN I don't see why it should be included (DADA2 is also unlikely to feature strongly on the differential diagnosis list). With the diagnostician in mind I would prefer to see the table comparing the clinical features of old vs. new PAN against those of GPA/MPA, keeping the description of VEXAS & DADA2 in the body of the text. However, this is just my personal preference and I would be happy for the article to be published without amendment.
I could not find any major problems with the review.
Round 2
Reviewer 1 Report
Comments and Suggestions for Authors
I agree with the changes made, which significantly improve the quality of the manuscript.